# Bisphenol A Removal Using Visible Light Driven Cu_2_O/PVDF Photocatalytic Dual Layer Hollow Fiber Membrane

**DOI:** 10.3390/membranes12020208

**Published:** 2022-02-10

**Authors:** Siti Hawa Mohamed Noor, Mohd Hafiz Dzarfan Othman, Watsa Khongnakorn, Oulavanh Sinsamphanh, Huda Abdullah, Mohd Hafiz Puteh, Tonni Agustiono Kurniawan, Hazirah Syahirah Zakria, Tijjani El-badawy, Ahmad Fauzi Ismail, Mukhlis A. Rahman, Juhana Jaafar

**Affiliations:** 1Advanced Membrane Technology Research Centre (AMTEC), School of Chemical and Energy Engineering, Universiti Teknologi Malaysia, Skudai 81310, Johor, Malaysia; sitihawamnoor@gmail.com (S.H.M.N.); mhafizputeh@utm.my (M.H.P.); hazirahzakria@gmail.com (H.S.Z.); el-badawy@graduate.utm.my (T.E.-b.); afauzi@utm.my (A.F.I.); mukhlis@petroleum.utm.my (M.A.R.); juhana@petroleum.utm.my (J.J.); 2Department of Civil and Environmental Engineering, Faculty of Engineering, Prince of Songkla University, Songkhla 90110, Thailand; watsa.k@psu.ac.th; 3Faculty of Environmental Science, Dongdok Campus, National University of Laos, Xaythany District, Vientiane 01080, Laos; oulavanhnoi@gmail.com; 4Department of Electrical, Electronic & Systems Engineering, Faculty of Engineering & Built Environment, The National University of Malaysia, Bangi 43600, Selangor, Malaysia; huda.abdullah@ukm.edu.my; 5School of Civil Engineering, Faculty of Engineering, Universiti Teknologi Malaysia, Skudai 81310, Johor, Malaysia; 6College of the Environment and Ecology, Xiamen University, Xiamen 361102, China; tonni@xmu.edu.cn

**Keywords:** bisphenol A, photocatalytic activity, visible light photocatalytic dual layer hollow fiber membrane

## Abstract

Bisphenol A (BPA) is amongst the endocrine disrupting compounds (EDCs) that cause illness to humans and in this work was removed using copper (I) oxide (Cu_2_O) visible light photocatalyst which has a narrow bandgap of 2.2 eV. This was done by embedding Cu_2_O into polyvinylidene fluoride (PVDF) membranes to generate a Cu_2_O/PVDF dual layer hollow fiber (DLHF) membrane using a co-extrusion technique. The initial ratio of 0.25 Cu_2_O/PVDF was used to study variation of the outer dope extrusion flowrate for 3 mL/min, 6 mL/min and 9 mL/min. Subsequently, the best flowrate was used to vary Cu_2_O/PVDF for 0.25, 0.50 and 0.75 with fixed outer dope extrusion flowrate. Under visible light irradiation, 10 mg/L of BPA was used to assess the membranes performance. The results show that the outer and inner layers of the membrane have finger-like structures, whereas the intermediate section of the membrane has a sponge-like structure. With high porosity up to 63.13%, the membrane is hydrophilic and exhibited high flux up to 13,891 L/m^2^h. The optimum photocatalytic membrane configuration is 0.50 Cu_2_O/PVDF DLHF membrane with 6 mL/min outer dope flowrate, which was able to remove 75% of 10 ppm BPA under visible light irradiation without copper leaching into the water sample.

## 1. Introduction

As global development increases rapidly, the usage of polycarbonate plastics, epoxy resins, personal care products as well as thermal paper and toys also increase extensively. These products contain bisphenol A (BPA) in their production [1]. Similarly, according to Corrales et al. [2], medical equipment, flame retardants, electronics, building materials and automobiles also contain BPA. Bisphenol A (BPA), or known scientifically as 2,2-bis-(4-hydroxyphenyl)propane (C_15_H_16_O_2_) consists of two phenol molecules bonded by a bridge of methyl and two groups of methyl [3]. BPA is one of the endocrine disrupting compounds and acts as a synthetic estrogenic hormone which can bind to estrogenic receptors and affects the estrogenic pathway. This compound can adversely affect humans resulting in infertility, male sexual dysfunction, polycystic ovary syndrome, miscarriages, premature deliveries, cardiovascular disease, liver failure, obesity, thyroid function and liver function [4]. This happens due to too much exposure towards BPA. The compound may leak from products or equipment that are composed of BPA into the environment or because of direct waste disposal into the environment. In recent years, synthetic hormones and endogenous compounds are frequently detected in surface waters, soils, sewages and even groundwaters. It gets into our drinking water after being exposed to heat, acid or base. Even as this compound exists at concentrations below 1 µg/L, it still harmful towards the aquatic system, animals and humans [5]. BPA itself can be found in Malaysia’s tap water in the range of 3.5–59.8 ng/L, bottled mineral water at around 3.3 ± 2.6 ng/L and 11.3 ± 5.3 ng/L for poorly stored bottled mineral water [6].

The World Health Organization stated in 2009 that there is no observed adverse effect for BPA if exposure level is 5 mg/kg of body weight/day [7]. Currently, a study reported that allowed concentration of BPA presence in water by international regulation is 0.05 ppm [8]. The lethal concentrations 50 (LC50) for BPA are 31 µM for 96 h, 47 µM for 72 h, 72 µM for 48 h and 70 µM for 24 h exposure [9]. The terminal half-life for BPA in the body is about 17.6 ± 7.69 h [10]. Thus, various types of treatment were developed to remove BPA from water in the environment including adsorption, nanofiltration, biological agents and emulsion liquid membrane [11,12,13,14]. However, there are limitations for these technologies, such as fouling for adsorption, nanofiltration is only applicable for certain particle size and not for large particle size and biological agents need long time duration to achieve optimum performance while emulsion liquid membranes have low membrane stability due to emulsification reaction [15,16,17].

To overcome these limitations, photocatalysis process is touted as a more promising method as it has its own self-cleaning properties, not only applicable for certain particle size, but is also a stable and time-efficient process. Photocatalysis is an acceleration of photoreactions. This method has been employed in the breakdown of organic pollutants, particularly in wastewater, hydrogen production, air purification and disinfection [18]. This treatment method is not only inexpensive, but it is also reusable, environmentally benign and capable of full decomposition when compared to other methods [19].

When a photon of light strikes the surface of a semiconductor, the photocatalysis process begins [18]. Light absorption at a wavelength higher than the catalyst’s bandgap generates electron–hole pairs on the catalyst. The conductive band has an electron (e^−^) and valence band having a positive hole (h^+^) [20]. The holes will oxidize the donor molecules of water (H_2_O), producing hydroxyl radicals in the process. The hydroxyl radicals, which have considerable oxidizing power, are responsible for the breakdown of contaminants. The electrons in the conduction band combine with oxygen and generate superoxide ions as a result of the reduction process. In photocatalysis, a redox reaction occurs, which is triggered by electrons [18].

UV light photocatalyst and visible light photocatalyst are two types of photocatalysts. UV light photocatalysts, such as titanium dioxide (TiO_2_), require the use of a UV lamp to activate, but visible light photocatalysts only require a standard visible lamp or sunshine to activate [18]. As a result, visible light photocatalysts are more cost-effective. Copper (I) oxide (Cu_2_O), often known as cuprous oxide, has a relatively low bandgap, ranging from 2.0 to 2.4 eV, which makes it a strong candidate for solar energy harvesting. This chemical is plentiful, non-toxic, simple to synthesize, effective in absorbing visible light and inexpensive [21]. Photocatalysts, DNA biosensors, sensors, lithium-ion batteries [21], cancer therapeutic agents, printed electronics, antimicrobial agents and catalysis are just a few of the applications for Cu_2_O [22].

Integrating photocatalysis into membrane technology is also a promising method to produce a good technique to remove BPA. Membrane technology is a separation technique that has the advantages of being simple to maintain, using few chemicals, producing high-quality water, having little environmental impact and being widely used in water purification. It is a promising method for treating wastewater [23]. In comparison to other membrane configurations, hollow fiber membrane provides lower maintenance costs and higher operational availability [24]. Besides, it can also withstand high pressure [25]. In addition to single layer hollow fiber membranes, co-extruded dual layer hollow fiber membranes are now regarded amongst the most promising membrane structures and have been explored extensively for water and wastewater treatment application [26].

Because two separate polymer dope solutions are utilized in fabrication and each layer is integrated together, the dual layer hollow fiber membrane has good flexibility. This reduces the individual materials’ weaknesses [27]. In comparison to single layer hollow fiber membranes, dual layer hollow fiber membranes require less materials and have better membrane performance [28]. Polyvinylidene fluoride (PVDF) is a polymer that is biocompatible, non-degradable, chemically resistant, odorless, non-toxic and has a low surface tension, making it easy to clean [29]. Microfiltration, membrane bioreactors, ultrafiltration, membrane distillation, biofuels recovery, lithium-ion battery separator, gas separation and stripping and pollution removal from water are some of the technologies using PVDF as a component. PVDF is a versatile polymer that is widely utilized in membrane technology as a primary component [30].

Integrating dual layer hollow fiber (DLHF) membrane technology with visible light photocatalyst producing photocatalytic DLHF membrane that is activated under visible light is a promising technique. There are few studies on Cu_2_O photocatalytic membrane. Cu_2_O thin film removed 87.6% of methylene blue, 22.7% of methyl orange and 63.6% of ciprofloxacin under visible light irradiation for 180 min [31]. Other than that, reduced graphene oxide/Cu_2_O/TiO_2_ membrane sheet removed of 100% of 10 ppm methyl orange and 70% of 10 ppm aniline under UV-vis irradiation in 120 min [32]. However, there is no study yet on BPA removal using Cu_2_O/PVDF DLHF membrane under visible light irradiation. Therefore, the aim of this work is to examine the influence of outer layer dope flowrate on the characteristics of the DLHF membrane fabricated via co-extrusion technique and to investigate the effect of Cu_2_O loading in the outer layer on properties of the DLHF membrane.

## 2. Materials and Methods

### 2.1. Materials

Polyvinylidene fluoride (PVDF, Kynar 760 Series-powder, Solvay Specialty Polymers France) (molecular weight: 441,000 by GPC) was used as polymer. Polyethylene glycol 6000 (PEG, Sigma Aldrich, Burlington, MA, USA) was used as pore former. Dimethylacetamide (DMAc, Sigma Aldrich, Burlington, MA, USA) was used as solvent and copper oxide (Cu_2_O, Sigma Aldrich, Burlington, MA, USA) as photocatalyst. Ethanol (EtOH, Hayman, Australia) was used for the post treatment process. Bisphenol A (BPA, Sigma Aldrich, Burlington, MA, USA) used as contaminant.

### 2.2. Fabrication of Cu_2_O/PVDF DLHF Membrane

The polymer dope composition for Cu_2_O/PVDF DLHF membrane fabrication is shown in Table 1. PVDF, PEG and Cu_2_O were dried for 24 h in a 50 °C oven to eliminate moisture. For inner layer dope, PEG at desired amount was dissolved by DMAc for 24 h using overhead stirrer (IKA RW 20 digital) at 600 rpm and followed by addition of PVDF into the mixture at the desired amount and continually stirred until it reached a homogenous state. For outer layer dope, Cu_2_O at desired amount was dissolved by DMAc using overhead stirrer (IKA RW 20 digital) at 600 rpm for 24 h and PVDF added at desired amount afterwards.

The inner layer dope was ultrasonicated and left overnight at room temperature to remove the air bubble in the dope and homogenize it. The outer layer dope was also sonicated for 30 min at 180 W to ensure the photocatalyst was well dispersed and to remove the air bubble. The membrane was fabricated using dry-jet wet co-extrusion spinning under conditions of 26 rpm inner dope flowrate, outer dope flowrate of 3 mL/min, 6 mL/min and 9 mL/min, 8 rpm of bore fluid flowrate, air gap of 100 mm and take up speed of 4 rpm spinning condition. The dope was extruded through a triple orifice spinneret forming dual layer hollow fiber membrane. The fabricated membrane underwent post treatment processing by soaking in a water bath for 24 h for residual solvent removal, immersion in 50:50 wt% ethanol for 1 h and finally immersion in >99 wt% ethanol for 1 h to prevent the membrane from shrinking. The membranes were subsequently air-dried for 3 days prior to storage in zipped plastic bags for further analysis. The neat PVDF membrane was fabricated with the same spinning condition and method but consisted of only inner layer dope.

### 2.3. Membrane Characterization

#### 2.3.1. Morphological Structure

Scanning electron microscopy (SEM; TM3000, Hitachi, Tokyo, Japan) analysis was used to analyze the morphological characteristics of the surface and cross-sectional area of the constructed Cu_2_O/PVDF dual layer hollow fiber membrane. Under vacuum, each sample was placed on a stub and coated with gold for 3 min. Each sample’s SEM image was captured at various magnifications.

#### 2.3.2. Crystalline Property

X-ray diffraction is a technique for determining the chemical composition, physical properties and crystallographic structure of a substance. X-ray diffraction (XRD, model: D5000, SIEMENS, Munich, Germany) was used to examine the crystallinity and phase identification of photocatalytic Cu_2_O/PVDF dual layer hollow fibers. The analysis took place at a voltage of 40 kV and a current of 30 mA. It likewise used CUK-radiation with a wavelength of 0.15418 nm and an angular incidence of 2 = 20–80° with a scan step speed of 1°/min at an angular incidence of 2 = 20–80°.

#### 2.3.3. Hydrophilicity

The degree of hydrophilicity of a photocatalytic DLHF membrane was determined using a contact angle goniometer (model: OCA 15EC, Dataphysics) and image processing software that was connected to the analyzer. A 2 µL drop of deionized water was placed onto the membrane surface and the liquid contact angle was carefully measured at 10 different spots.

#### 2.3.4. Water Flux

A U-shaped membrane module filtering equipment was used to analyze pure water flux. Three DLHF membranes were grouped into a bunch of fibers with a length of 10 cm and were placed in the filtering module. The water flux was measured using an outside-in arrangement in cross flow mode. In flux rate analysis, distilled water was used as the permeating substance. The permeability of the membrane can be determined in this study by applying varying pressures to different membrane samples. Zero (0) bar of pressure was applied on the membrane for 10 min to compact the membrane and achieve a steady flux. Thereafter, the flow was measured, and flux was determined using Equations (1) and (2): (1)$$F=\frac{V}{A\times t}$$
(2)A=πdoL
where F is the membrane flux (L/m^2^h), V is the permeate volume (L) at time t (min), A is the membrane filtration area (m^2^), do is the hollow fiber’s outer diameter (cm) and L is the length of the hollow fibers (cm).

#### 2.3.5. Porosity

Gravimetric analysis was used to determine the membrane’s porosity as reported by Gonzales et al. [33]. Dry membranes were weighed and then soaked in water at room temperature for 24 h. Subsequently, the moist membranes were reweighed, and the porosity was determined using Equation (3):(3)$$\varepsilon =\frac{\frac{M2-M1}{\rho w}}{\frac{M2-M1}{\rho w}+\frac{M2}{\rho p}}$$
where *M*1 represents the weight of the membrane before it is soaked in water, *M*2 represents the weight of the soaked membrane, ρ*w* represents the density of water at room temperature and ρ*p* represents the density of the polymer.

#### 2.3.6. Surface Roughness

Atomic force microscopy (AFM) is a technique that can be used in air, liquid or vacuum to analyze a surface area down to atomic precision to obtain a topographic image with extremely high resolution. AFM was used to examine the surface of the DLHF membrane (AFM; model SE-100, Park System, Suwon, Korea). The sample was scanned using tapping mode after cutting the membrane into 5 cm pieces with 20 m × 20 m regions.

### 2.4. Performance Evaluation

A submerged photocatalytic system was designed to test the photocatalytic efficacy of the Cu_2_O/PVDF DLHF membrane in eliminating BPA from wastewater. Ten strands of DLHF membranes were strung together in a U-shape with lengths of approximately 23 cm. Hollow fiber membranes were potted in PVC tubes with epoxy resin (E-30CL Loctite Corporation, Westlake, OH, USA) and allowed to set at room temperature. The membranes were then inserted into the PVC adaptor. The system depicted in Figure 1 included a visible light bulb (LED flood light; model: IP65, 100 W), which is an essential component of the system since the photocatalyst activates and reacts well when exposed to visible light. A magnetic stirrer was inserted at the bottom of the beaker to guarantee that the solution was homogeneous, and that the pollutant concentration was uniform so that it could be degraded. Membrane modules were submerged in the beaker. To collect permeate, a peristaltic pump was attached to the system at a pressure of 0.05 Mpa. Furthermore, 10 mg/L of BPA was prepared and poured into the beaker. For 30 min, the BPA solution was kept in a dark chamber to achieve adsorption/desorption equilibrium. Following that, a blank sample of 10 mL of BPA solution was collected, as well as the initial concentration (Co) of BPA. The solution was then exposed to visible light for 30 min before aliquoting another 10 mL and repeating the process every 30 min until 360 min had passed.

To determine the presence of BPA in the water sample, the aliquoted samples were analyzed using a UV-vis spectrophotometer (DR5000, PerkinElmer, Llantrisant, Wales, UK). The wavelength used was 230 nm. The percentage degradation of BPA was then determined using Equation (4) and the absorbance result [34]:(4)$$\mathrm{Degradation}\;\mathrm{of}\;\mathrm{organic}\;\mathrm{contaminants}=\frac{\mathrm{Co}-\mathrm{Ct}}{\mathrm{Ct}}\times 100$$
where Co is the initial concentration at t = 0 and Ct is the concentration at collection time in each cycle as present in Section 2.4.

### 2.5. Leaching Compound Detection

Copper leaching analysis of the degraded BPA water sample using Cu_2_O/PVDF DLHF membrane was analyzed by ICP-OES (inductively coupled plasma—optical emission spectrometry, model: 710, Agilent Technologies, Santa Clara, CA, USA). The treated BPA water samples or permeate was used as a sample to guarantee that no copper leaching from the Cu_2_O/PVDF DLHF membrane occurred during the treatment process. All selected spectral lines for ICP-OES were free of interference. The sample and calibration solutions were prepared in the same media, which reduced background emission.

## 3. Results and Discussion

### 3.1. The Effect of Different Outer Dope Flowrate

#### 3.1.1. Morphological Structure

SEM analysis was performed on 0.25 Cu_2_O/PVDF DLHF membranes with three distinct outer dope flowrates of 3 mL/min, 6 mL/min and 9 mL/min. The SEM pictures of the examined samples are shown in Figure 2. Overall, the cross-sectional scans revealed that the membrane had a finger-like structure at the inner and outer ends, and a sponge-like structure in the center which is also known as a sandwich-like structure. The instability of the suspension-coagulant contact led to the sandwich-like structure. Sponge like structure formation was caused by the slow precipitation rate, whereas the formation of the finger-like or porous structure was caused by the fast precipitation rate caused by the intrusion of bore fluid and coagulation bath into the inner and outer surfaces, respectively [35,36]. Higher flowrates of the outer layer dope caused photocatalyst aggregation, which appears as white spots in the membrane’s outer layer and clogs the pores. A higher outer dope flowrate results in a thicker outer layer. The faster the outer dope flow rate, the more pressure was applied to the inner layer, resulting in a suppressed finger-like shape.

In comparison to outer dope flowrate membranes, the 0.25 Cu_2_O/PVDF DLHF membrane with 9 mL/min outer dope flowrate has the longest outer layer finger-like structure (Figure 2). When the outer dope flowrate is increased from 3 to 6 mL/min, the pore size increases. The pore size for 9 mL/min, however, reduced due to photocatalyst aggregation on the membrane’s surface, which clogged the pores. As a result, with a 0.25 Cu_2_O/PVDF DLHF membrane, a flowrate of 6 mL/min of outer dope is the optimum flowrate for the membrane to function well.

The thickness of the membrane was also measured using the ImageJ application and is shown in Table 2. Furthermore, 0.25 Cu_2_O/PVDF DLHF membranes have a 13.05 nm outer layer thickness for a 3 mL/min outer layer dope flowrate. Furthermore, 0.25 Cu_2_O/PVDF DLHF membranes have a 21.35 nm outer layer thickness for a 6 mL/min outer layer dope flowrate. Finally, the outer layer dope flowrate of 0.25 Cu_2_O/PVDF DLHF membranes is 9 mL/min, with an outer layer thickness of 89.35 nm. The thickness of the outer layer grows as the outer layer dope flowrate increases. As a result, there are more Cu_2_O particles in the membrane’s outer layer.

Cu_2_O/PVDF DLHF membrane physical configuration is similar to results from the work of Kamaludin et al. [37] and Dzinun et al. [38]. Both studies focus on the fabrication of a dual layer hollow fiber membrane utilizing the co-extrusion process. They have similar membrane features to the Cu_2_O/PVDF DLHF membrane, which has a sponge-like structure in the middle layer and a finger-like structure in the inner and outer layers. Similar to what they have reported, the photocatalyst is retained in the outer layer of the Cu_2_O/PVDF DLHF membrane. The addition of PEG 6000 to the inner layer dope solution causes the creation of large finger-like structures in the Cu_2_O/PVDF DLHF membrane. The pore former is added to make the membrane more porous, allowing for higher water flux and solute rejection [39]. The addition of Cu_2_O to the PVDF membrane’s outer layer improves the membrane’s hydrophilicity [40]. When the membrane’s outer layer is hydrophilic, it allows a lot of water to flow into the pore. The amount of Cu_2_O in the membrane increases its hydrophilicity and pore size [39]. The Cu_2_O/PVDF DLHF membrane’s inner layer has a sponge-like structure. The sponge-like structure is a network of interconnected nodes. The structure is ideal for a high mechanical strength membrane [41]. During direct contact membrane distillation trials, the sponge-like structure also improved vapor permeability [42].

#### 3.1.2. Crystalline Property

Figure 3 shows the XRD peaks of the Cu_2_O/PVDF DLHF membrane. The figure shows that the membrane is comprised of PVDF and Cu_2_O, with peaks at 2θ = 18.43°, 20.22°, 29.47°, 36.367°, 42.24°, 61.31°, 73.49° and 77.28°. The interaction between PVDF and Cu_2_O has transitioned from phase to phase in the membrane, as indicated by the crystal structure peaks [43]. PVDF membrane XRD peaks at 18.2° and 20.2°, according to Wu et al. [44], agree with Cu_2_O/PVDF DLHF membrane XRD peaks for PVDF crystal structure. Cu_2_O has peaks at 29.68°, 36.68°, 42.60°, 61.76°, 73.98° and 77.80° indicated as (110), (111), (200), (220) and (311) diffraction planes. According to Hssi et al. [45] and He et al. [46], indicating that the XRD result for Cu_2_O/PVDF DLHF membrane has the expected crystal peak for Cu_2_O.

#### 3.1.3. Hydrophilicity

Figure 4 shows the hydrophilicity of Cu_2_O/PVDF DLHF membrane. With a 3 mL/min outer dope flowrate, the mean contact angle reading for 0.25 Cu_2_O/PVDF DLHF membrane is 62.39°, 61.27° for 6 mL/min outer dope flowrate and 59.32° for 9 mL/min outer dope flowrate. The mean contact angle reading decreases when the outer dope flowrate increases for 0.25 Cu_2_O/PVDF DLHF membrane. This indicates that adding a small quantity of Cu_2_O to the membrane makes it more hydrophilic, whereas a pure PVDF membrane is hydrophobic. PVDF membranes have a very high contact angle of around 92.19°, which is classified as hydrophobic, and after a few treatments, they can attain a contact angle of 120° [47].

#### 3.1.4. Porosity

Figure 5 shows the porosity of Cu_2_O/PVDF DLHF membrane. For 3 mL/min outer dope flowrate, the 0.25 Cu_2_O/PVDF DLHF membrane has 41.84% porosity, while outer dope flow rates of 6 mL/min and 9 mL/min recorded porosities of 63.13% and 42.76%, respectively. The porosity of the 0.25 Cu_2_O/PVDF DLHF membrane increased with increase of flow rate from 3 to 6 mL/min and decreased when the outer dope flowrate climbed to 9 mL/min. The pore size trend determines the porosity trend as both are seen to be in sync. The pore size increased and then decreased at 9 mL/min outer dope flowrate, similar to the porosity pattern, from 3 to 6 mL/min of outer dope flowrate.

Increase in membrane porosity is governed by increase in membrane pore size [48]. PVDF membranes for microfiltration and ultrafiltration have wide pore sizes ranging from 150 to 450 nm [49]. The PVDF membrane has a porosity of more than 80%. The study backs up the claim that increasing pore size increases membrane porosity, and that the membrane’s application is determined by pore size selectivity, which can be microfiltration, ultrafiltration, nanofiltration or reverse osmosis.

#### 3.1.5. Water Flux

Water flux of Cu_2_O/PVDF DLHF membrane is shown in Figure 6. Furthermore, 0.25 Cu_2_O/PVDF DLHF membrane flux values are 2278.62 L/m^2^h for 3 mL/min outer dope flowrate, 4919.02 L/m^2^h for 6 mL/min outer dope flowrate and 421.37 L/m^2^h for 9 mL/min outer dope flowrate. The results reveal that a 6 mL/min outer dope flowrate performs well on water flux readings. Membrane water flow is influenced by membrane pore size and porosity. High membrane water flow is aided by large pore size and porosity [50]. Furthermore, 9 mL/min outer dope flowrate has the least flux. This is not unconnected to its small pore size and poor porosity. In addition, it can also be related to the membrane thickness as shown in Table 2. Furthermore, 9 mL/min outer dope flowrate has the maximum outer layer membrane thickness. Thus, increasing the membrane thickness and increasing the filtration pathway of the membrane. As is established, thicker membrane has lower flux [51].

As a result, the best flowrate for the subsequent aim (the effect of different Cu_2_O/PVDF ratio loading) was determined to be 6 mL/min outer dope flowrate. This is because it produced the largest pore size, highest porosity and highest water flux. We can conclude that high flow rate is induced by the long finger-like structure and high porosity resulting in high water flux.

### 3.2. The Effect of Different Cu_2_O/PVDF Ratio Loading

#### 3.2.1. Morphological Structure

Table 3 shows the morphological structure for neat PVDF membrane, 0.25, 0.50 and 0.75 Cu_2_O/PVDF DLHF membranes at an outer dope flow rate of 6 mL/min. The pore size for neat PVDF membrane is 91 nm. Meanwhile, the pores size of the Cu_2_O/PVDF DLHF membrane increases from 100.88 to 129.34 nm with a corresponding increase of Cu_2_O ratio from 0.25 to 0.50, and subsequently reduces to 96.00 nm when the Cu_2_O ratio reaches 0.75. The deposition of copper particles was determined by EDX analysis on neat PVDF membrane, 0.25, 0.50 and 0.75 Cu_2_O/PVDF DLHF membranes with a 6 mL/min outer dope flowrate. Table 3 shows that copper particles are only present in the membrane’s outer layer and do not penetrate the inner layer and the absence of copper particle in neat PVDF membrane. When the ratio of Cu_2_O to PVDF is increased from 0.25 to 0.75, the number of copper particles that are colored red increases. Furthermore, 1.5 wt%, 2.6 wt% and 8.0 wt% copper was detected in the 0.25 Cu_2_O/PVDF, 0.5 Cu_2_O/PVDF and 0.75 Cu_2_O/PVDF DLHF membranes, respectively. In the same vein, the outer layer thickness of the 0.25 Cu_2_O/PVDF, 0.50 Cu_2_O/PVDF and 0.75 Cu_2_O/PVDF DLHF membranes were 22.04 nm, 52.20 nm and 85.13 nm, respectively.

The Cu_2_O/PVDF DLHF membrane with a composition of 0.75 Cu_2_O/PVDF had the highest Cu_2_O/PVDF ratio. As a result, the photocatalyst agglomerated in the outside dope region, resulting in many white spots. The outer finger-like structure is longer than the inner finger-like structure, and the inner finger-like structure has been repressed. This is due to the lower concentration of PVDF in the outer dope’s composition, the finger-like structure is longer than 0.25 Cu_2_O/PVDF DLHF membrane and 0.50 Cu_2_O/PVDF DLHF membrane.

The pore size increase as the addition of Cu_2_O into neat PVDF membrane will thus increase the membrane hydrophilicity and permeability. Cu_2_O agglomeration occurred as the pore size decreased at the maximum ratio of Cu_2_O to PVDF (0.75). Because of the strong surface forces between the Cu_2_O nanoparticles and the DMAc solvent, agglomeration set in [52]. When DMAc interacts with Cu_2_O and causes the particle to clump during the phase inversion process, colloidal instability is created. The agglomeration of nanoparticles is a major issue for membrane performance, resulting in reduced water flow and reduced mechanical strength [53]. It obstructs pores, reducing flow and diminishing membrane permeability. Furthermore, agglomeration increases particle size, reducing the overall surface area of the particle and affecting photocatalytic efficacy [54].

A larger Cu_2_O/PVDF ratio increases the thickness of the membrane’s outer layer. According to Bensouici et al. [55], increasing the photocatalyst coating to nine layers of photocatalyst coating, increases the thickness of the photocatalytic layer. Cu_2_O penetration into the inner layer is limited, which permits the majority of Cu_2_O to be exposed to the light source. More active sites for the photocatalytic process occur when the surface area of Cu_2_O is exposed to light. Co-extrusion, a single-step membrane manufacturing technology, effectively produced well-distributed Cu_2_O on the outer layer membrane.

#### 3.2.2. Hydrophilicity

Cu_2_O/PVDF DLHF membrane’s water contact angle (Figure 7) varies with the three Cu_2_O/PVDF ratios of 0.25, 0.50 and 0.75. The outer dope flowrate for each membrane composition is 6 mL/min. The mean contact angle reading for a 0.25 Cu_2_O/PVDF DLHF membrane is 61.27° ± 1.66°. Meanwhile, the average contact angle reading for a 0.50 Cu_2_O/PVDF DLHF membrane is 58.90° ± 1.72°, while the average contact angle for a 0.75 Cu_2_O/PVDF DLHF membrane is 61.16° ± 0.59°.

The mean contact angle measurement decreases when the outer dope flowrate increases for 0.25 Cu_2_O/PVDF DLHF membrane and 0.50 Cu_2_O/PVDF DLHF membrane. Whereas the contact angle for neat PVDF membrane is 93.58° ± 1.27°. This shows that adding a small quantity of Cu_2_O to the membrane makes it more hydrophilic, whereas a pure PVDF membrane is hydrophobic. The addition of Cu_2_O to a membrane increases the membrane’s hydrophilicity and pore size [38]. However, the contact angle of the 0.75 Cu_2_O/PVDF DLHF membrane increased, making it less hydrophilic than the others. This was caused by Cu_2_O aggregation at the outer layer, which clogged the pore openings since it was the smallest membrane. In Ahmad et al.’s [56] work, ZnO agglomerated on the polyethersulfone membrane after the concentration was raised over 1.0 wt%. Consequently, the pores became clogged, and the pore size decreased. Because pore size affects water diffusion, a smaller pore size means a larger contact angle [50].

#### 3.2.3. Porosity

Figure 8 shows the porosity of neat PVDF membrane, Cu_2_O/PVDF DLHF membranes with membranes of 0.25, 0.5 and 0.75 Cu_2_O/PVDF ratios. The neat PVDF membrane has 35.1% ± 1.73 porosity. While the porosity of a 0.25 Cu_2_O/PVDF DLHF membrane with an outer dope flowrate of 6 mL/min is 63.13% ± 5.0483. With the same outer dope flowrate, 0.50 Cu_2_O/PVDF DLHF membrane recorded a porosity of 44.15% ± 2.9590 while that of 0.75 Cu_2_O/PVDF DLHF membrane was 42.37% ± 6.2498. The addition of Cu_2_O into the membrane increase the porosity as neat PVDF membrane, which has smallest pore size, has the lowest porosity. According to these findings, the Cu_2_O/PVDF DLHF membrane has a high porosity. The trend, however, is downward. The 0.5 and 0.75 Cu_2_O/PVDF DLHF membranes’ porosity decreased more due to increased photocatalyst loading than the 0.25 Cu_2_O/PVDF DLHF membrane. The membrane porosity is affected by the amount of photocatalyst loaded. The membrane porosity also correlated with the membrane thickness as it affects the membrane transport and permeability. As the ratio increase from 0.25 to 0.75, the thickness of the membrane increases and porosity decreases. Thinner membrane at 0.25 ratio with resulting higher diffusion and higher porosity [57].

#### 3.2.4. Water Flux

Figure 9 shows the water flux readings for Cu_2_O/PVDF DLHF membranes with 0.25, 0.5 and 0.75 Cu_2_O/PVDF at a 6 mL/min outer dope flowrate. Furthermore, 0.25 Cu_2_O/PVDF DLHF membrane reached flux value of 4919.02 ± 98.91 L/m^2^h while the flux for 0.50 Cu_2_O/PVDF DLHF membrane was 13,890.99 ± 164.96 L/m^2^h. Finally, the flux value for 0.75 Cu_2_O/PVDF DLHF membrane with the same outer dope flow rate of 6 mL/min was 12,534.44 ± 87.08 L/m^2^h. Meanwhile, the flux value for neat PVDF membrane is 33.25 ± 1.29 L/m^2^h which is much lower compared to Cu_2_O/PVDF DLHF membranes. Thus, proving that addition of Cu_2_O photocatalyst increases the membrane permeability and water flux.

Based on the findings, it was determined that increasing the photocatalyst loading might have enhanced the membrane’s water flow. However, once the membrane has achieved its maximum photocatalyst concentration, adding more photocatalyst would reduce the water flux performance. The water flow of the 0.75 Cu_2_O/PVDF DLHF membrane began to decrease, indicating that the amount of photocatalyst is too large and compacted for the membrane to function properly. This can also happen when photocatalyst clumps together on the membrane’s outer layer, clogging holes and lowering membrane permeability [58]. As a result, the ideal ratio for the membrane to perform optimally is the 0.5 Cu_2_O/PVDF DLHF membrane.

Few studies have shown that adding photocatalyst enhances membrane water flow and hydrophilicity. One such research looked at graphene oxide and graphene oxide-titanium dioxide mixtures in polysulfone membranes. By combining polysulfone membranes with other semiconductors, the contact angle of the membrane reduced, making it more hydrophilic and permeable to high water flux [59]. Another work reported increasing the water flow by incorporating graphene oxide into a PVDF membrane and varying the pH from 4 to 10. At pH 10, the highest flow of 290 L/m^2^h was achieved [60]. Besides, Zr-based metal organic frameworks (MOF) were integrated onto spinel magnetic nanoparticles (SMNP) to improve the hydrophilicity of the SMNPs and MOFs [36].

#### 3.2.5. Surface Roughness

Table 4 shows the surface roughness of the neat PVDF membrane and Cu_2_O/PVDF DLHF membrane. With the same outer dope flowrate of 6 mL/min, three types of Cu_2_O/PVDF DLHF membranes were tested. Total surface roughness of neat PVDF membrane is 6.21 nm, 0.25 Cu_2_O/PVDF DLHF membrane was 7.13 nm, while that of 0.50 Cu_2_O/PVDF DLHF membrane was 7.51 nm (Ra). Meanwhile, the 0.75 Cu_2_O/PVDF DLHF membrane had the highest surface roughness value at 32.64 nm.

Increase of surface roughness causes increase of contact angle [61]. However, the concept is not applicable in this study. The increase of surface roughness decreases the contact angle. This situation was also reported by Junkar’s [62] research study where the contact angle reading also decreased as surface roughness increased. According to Busscher et al. [63], surface roughness lower than 100 nm will not affecting the contact angle reading. Thus, influence of surface roughness on contact angle is negligible in this study. In this study, the factors that influence the membrane contact angle are the pore size and the addition of hydrophilic material into the membrane composition [64].

Due to Cu_2_O agglomeration, the 0.75 Cu_2_O/PVDF DLHF membrane exhibits the maximum surface roughness. According to Ahmad et al. [56], the aggregation of nanoparticles on the membrane’s outer layer enhances the membrane’s surface roughness. Although the surface roughness is the maximum, aggregation has plugged the pore, resulting in a lower pore size and reduced membrane performance. We can then conclude that 0.5 Cu_2_O/PVDF DLHF membrane has the optimum surface roughness compared to the other two other ratio compositions.

### 3.3. Photocatalytic Activity

Adsorption-desorption equilibrium testing was performed on the membrane, which required 50 min of adsorption-desorption equilibrium before light irradiation could begin. Furthermore, 6 mL/min outer dope flowrate for all three ratios of membrane was further studied for its photocatalytic activity in degrading 10 mg/L of BPA, as shown in Figure 10. In comparison to 0.25 and 0.75 ratios, the 0.5 ratio Cu_2_O/PVDF DLHF membrane exhibits the best photocatalytic performance in degrading BPA. In 330 min under visible light, a 0.5 ratio Cu_2_O/PVDF DLHF membrane decomposed 75% of 10 mg/L. Meanwhile, the 0.25 ratio Cu_2_O/PVDF DLHF membrane degraded 69.57% of 10 mg/L BPA, while the 0.75 ratio Cu_2_O/PVDF DLHF membrane only degraded 55.55%. The outer layer of the Cu_2_O/PVDF DLHF membrane serves as a photocatalytic activity degradation site, while the inner layer serves as a separation barrier. As the photocatalyst is exposed on the outer layer of the membrane and easily activated with visible light, the photocatalytic process appears promising for Cu_2_O/PVDF DLHF membrane.

In comparison to the other membranes, the 0.75 ratio Cu_2_O/PVDF DLHF membrane exhibited the lowest performance due to photocatalyst agglomeration on the outer layer. Agglomeration of photocatalysts reduces the total surface area and hence the exposed surface area towards the light source [65]. Agglomeration of photocatalysts lowered membrane porosity and the overall active site surface area available for photocatalytic degradation [66]. It also enhanced surface roughness and blocked pores, resulting in a low water flux membrane. As a result of the high ratio of Cu_2_O to PVDF, the photocatalytic activity of the 0.75 Cu_2_O/PVDF DLHF membrane is low.

The performance of the 0.25 Cu_2_O/PVDF DLHF membrane is greater than that of the 0.75 Cu_2_O/PVDF DLHF membrane, but it is lower than that of the 0.50 Cu_2_O/PVDF DLHF membrane. This occurred because the 0.25 Cu_2_O/PVDF DLHF membrane contained less Cu_2_O as a photocatalyst than the 0.5 Cu_2_O/PVDF DLHF membrane. The 0.50 Cu_2_O/PVDF DLHF membrane displays the best performance in decomposing 10 ppm BPA.

When compared to the work of Kamaludin et al. [67], the N-doped TiO_2_/PVDF DLHF membrane eliminated 81.6 percent of BPA with a starting concentration of 5 ppm after 360 min of visible light irradiation. Meanwhile, the 0.50 Cu_2_O/PVDF DLHF membrane from this work removed 75% of BPA at a concentration of 10 mg/L, more than doubling the concentration of 5 mg/L, demonstrating that the 0.50 Cu_2_O/PVDF DLHF membrane has a high removal efficiency.

Similarly, a kaolin-based Ag@TiO_2_ hollow fiber membrane was used to remove a BPA content of 10 mg/L. Under visible light irradiation, the photocatalyst successfully eliminated 90.51% of the BPA in 180 min [68]. The performance is superior to that of the Cu_2_O/PVDF DLHF membrane. However, the fabrication process of the kaolin-based Ag@TiO_2_ hollow fiber membrane is rather cumbersome and involves many steps. As a ceramic membrane, kaolin-based Ag@TiO_2_ hollow fiber membrane also has the problem of the brittleness of ceramic materials, which makes the forming and assembly of the membrane more difficult compared to Cu_2_O/PVDF DLHF membrane, which is a polymeric membrane [69].

In addition, photocatalytic BPA elimination was detected using ZnO, TiO_2_ and SnO_2_ after three hours of UV light irradiation. The original concentration of 25 mg/L BPA was reduced by 89% with 0.1% (*w*/*w*) ZnO, 46% with TiO_2_ and 37% with SnO_2_. After five hours, ZnO had removed 98% of the 25 mg/L BPA, TiO_2_ had removed 65% of the BPA and SnO_2_ had removed 48% of the BPA [70]. In the presence of hydrogen peroxide at pH 8, biohybrid CuNPs@CALB-3, which contained Cu_2_O nanoparticles of roughly 10 nm size, demonstrated outstanding catalytic performance, eliminating more than 95% BPA at 45 ppm in an aqueous solution in 20 min using 1.5 g/L of a catalyst [8]. Each catalyst has good photocatalytic activity; however, these catalysts require UV light to activate, whereas the Cu_2_O/PVDF DLHF membrane can be activated with an inside house lamp and sunlight.

WO_3_ was integrated with molybdenum disulfide (MoS_2_) and silver (Ag) forming WO_3_@MoS_2_/Ag hollow nanotubes with mass ratio Ag to HW@M 7% and combined with peroxymonosulfate. The WO_3_@MoS_2_/Ag hollow nanotubes were observed to degrade 10 mg/L of BPA under visible light irradiation with initial nanotubes concentration of 40 mg. The nanotube successfully removed 92.51% of the BPA in 140 min and pure WO_3_, also tested on BPA, only removed 10.55% [71]. The nanotubes have a complicated preparation process while pure WO_3_ tends to cause fouling. Cu_2_O was integrated into a visible light/Cu_2_O/H_2_O_2_ system and it totally degraded 10 mg/L BPA with an initial dose of Cu_2_O of 0.112 g which is good but also tends to cause fouling and requires secondary treatment for recovery [72]. The comparison between Cu_2_O/PVDF DLHF membrane and other photocatalysts is summarized as in Table 5.

Based on these findings, it can be stated that the Cu_2_O/PVDF DLHF membrane is one of the most promising when compared to other photocatalyst membranes, since the other photocatalyst membranes eliminated smaller concentrations than the Cu_2_O/PVDF DLHF membrane. In addition, preparing the membrane is easier than preparing the other types of photocatalyst membranes. Furthermore, other types of photocatalyst exist in suspension form that require additional treatment after the operation, but the Cu_2_O/PVDF DLHF membrane does not require any treatment after the photocatalysis process.

### 3.4. Leaching Test

The leaching of copper element from the membrane into the treated BPA water sample was investigated during the treatment process. There was no trace of copper in the water sample, indicating that there is no copper leaching from the membrane and that the co-extrusion approach for fabricating Cu_2_O/PVDF dual layer hollow fiber membrane is the best way to ensure that the photocatalytic treatment procedure is safe. It is important to mitigate copper leaching in order to avoid disrupting ecosystem balance and to ensure that the membrane could work ideally without losing any photocatalyst particles that could impact its photocatalytic activity. Copper leaching is common for Cu_2_O/ZnO matrices [73]. Hence, the Cu_2_O/PVDF DLHF membrane is an ideal membrane with no copper leaching.

## 4. Conclusions

As part of the efforts to degrade and eliminate EDC waste, BPA, this study developed a Cu_2_O composited PVDF dual layer hollow fiber membrane. To identify the ideal concentration for Cu_2_O in the photocatalytic Cu_2_O/PVDF DLHF membrane, the photocatalyst/polymer ratio in the composition was varied between 0.25, 0.50 and 0.75, while the outer dope flowrate was modified between 3, 6 and 9 mL/min. SEM, XRD and AFM were used to analyze and evaluate the structural and chemical properties of the membranes. Meanwhile, the membranes’ physical features were put to the test, including contact angle, porosity and flow rate analysis. Under visible light irradiation, the BPA is successfully eliminated by the 0.50 Cu_2_O/PVDF DLHF photocatalytic membrane with 6 mL/min outer dope flowrate by about 75% of the 10 ppm BPA in 360 min, without the copper element being leached.

## Figures and Tables

**Figure 1 membranes-12-00208-f001:**
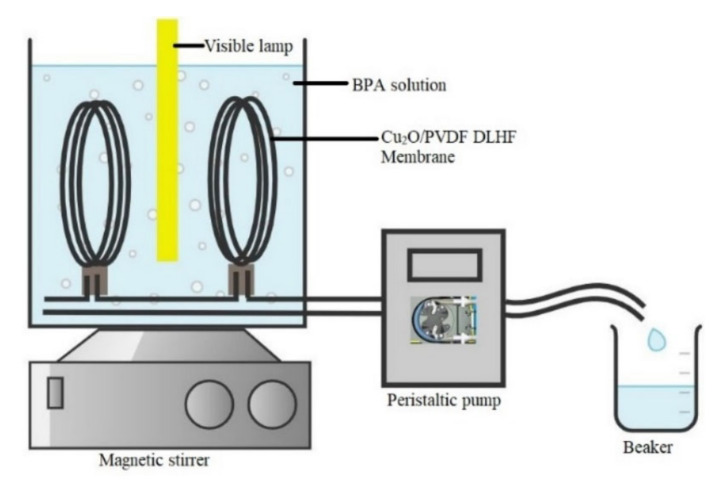
Schematic diagram of pilot-scale submerged photocatalytic membrane reactor.

**Figure 2 membranes-12-00208-f002:**
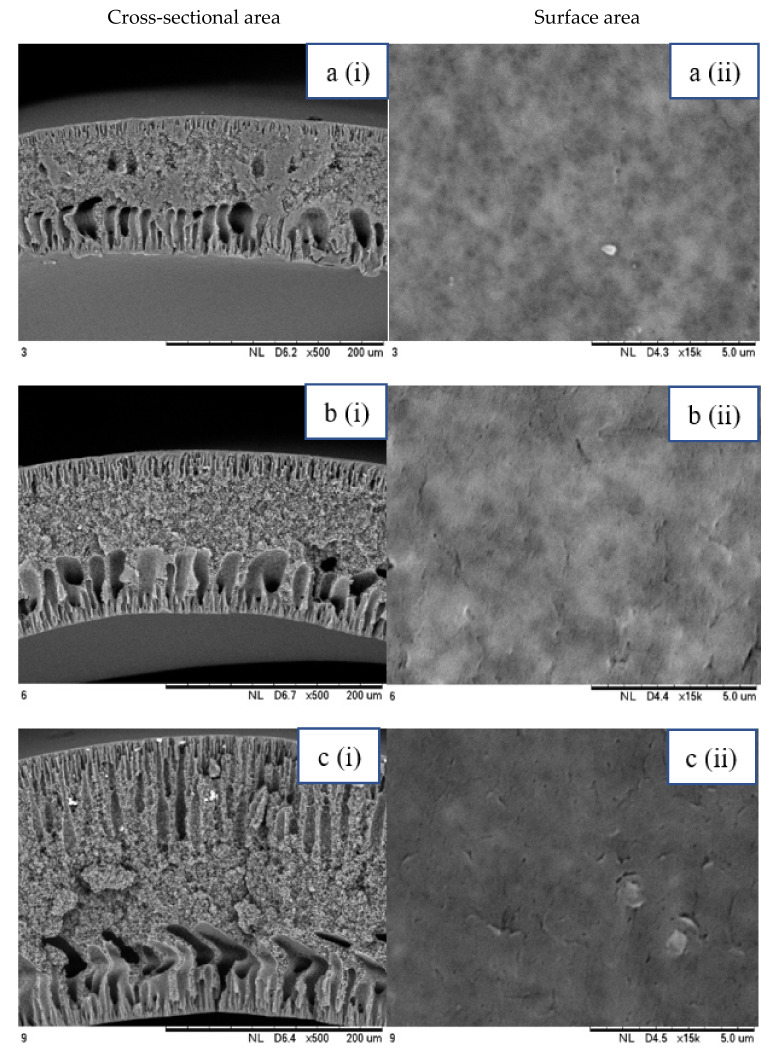
SEM image of 0.25 Cu_2_O/PVDF DLHF membrane with different outer dope flowrates: (**a**) 3 mL/min, (**b**) 6 mL/min, (**c**) 9 mL/min. Cross-sectional area (i); Surface area (ii).

**Figure 3 membranes-12-00208-f003:**
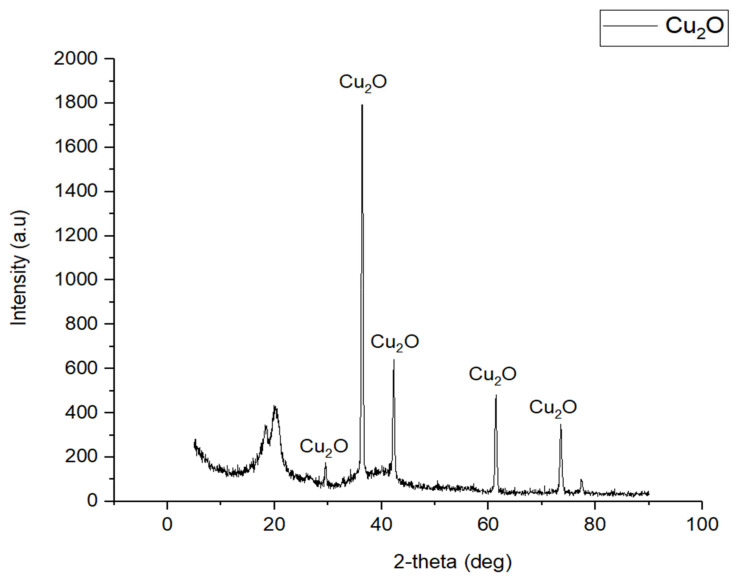
XRD graph for 0.5 Cu_2_O/PVDF DLHF membrane with 6 mL/min outer dope extrusion flowrate.

**Figure 4 membranes-12-00208-f004:**
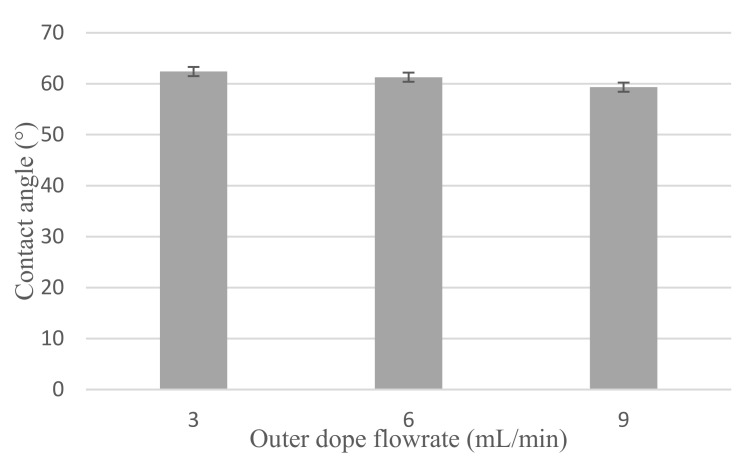
Contact angle for 0.25 Cu_2_O/PVDF DLHF membrane for 3 mL/min, 6 mL/min and 9 mL/min outer dope flowrate.

**Figure 5 membranes-12-00208-f005:**
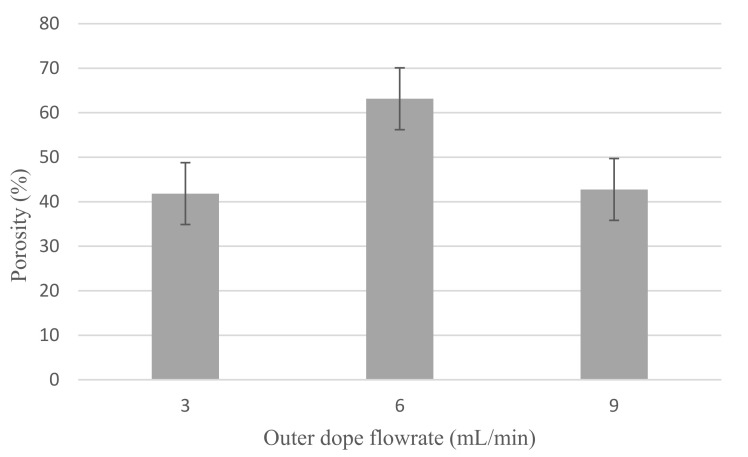
Porosity of 0.25 Cu_2_O/PVDF DLHF membrane for 3 mL/min, 6 mL/min and 9 mL/min outer dope flowrate.

**Figure 6 membranes-12-00208-f006:**
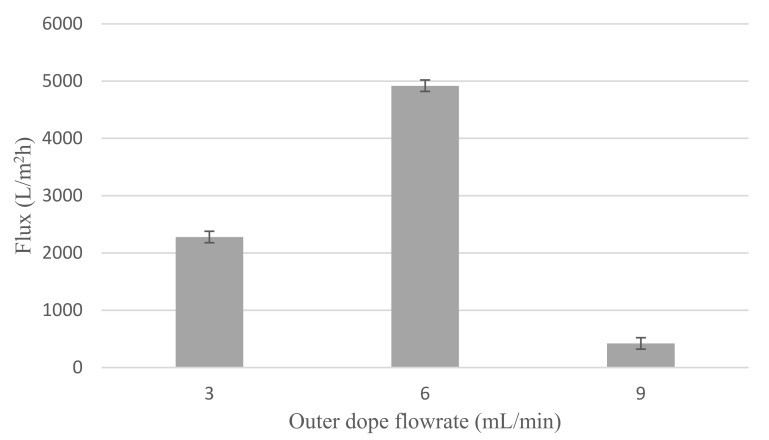
Water flux reading of 0.25 Cu_2_O/PVDF DLHF membrane for 3 mL/min, 6 mL/min and 9 mL/min outer dope flowrate.

**Figure 7 membranes-12-00208-f007:**
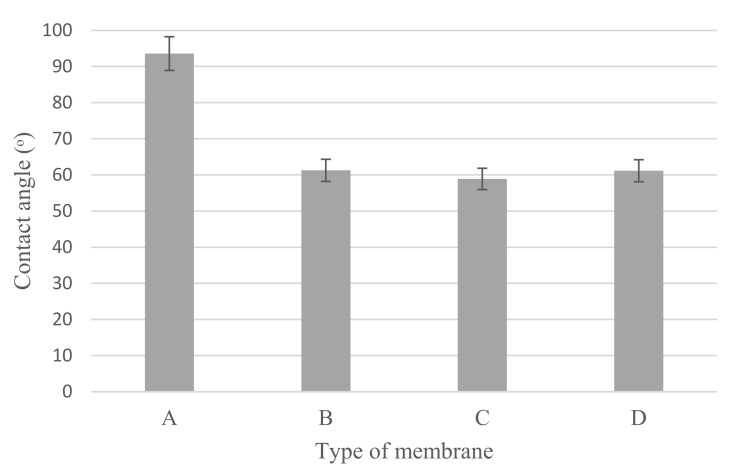
Contact angle of (A) neat PVDF membrane, (B) 0.25 Cu_2_O/PVDF DLHF membranes, (C) 0.5 Cu_2_O/PVDF DLHF membranes and (D) 0.75 Cu_2_O/PVDF DLHF membranes with outer dope flowrate 6 mL/min.

**Figure 8 membranes-12-00208-f008:**
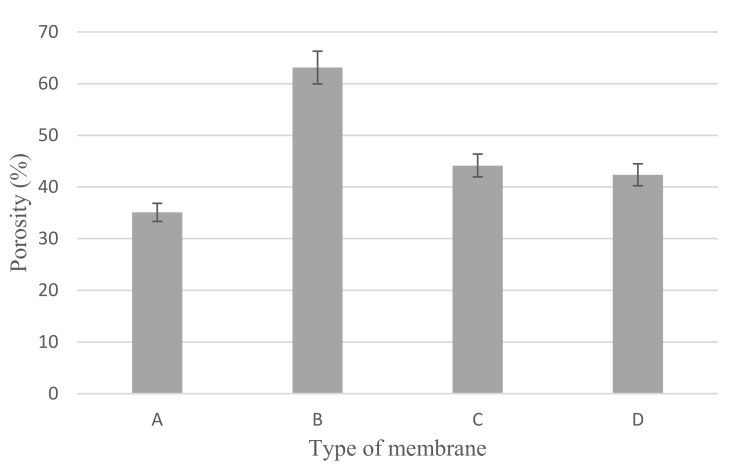
Porosity of (A) neat PVDF membrane, (B) 0.25 Cu_2_O/PVDF DLHF membranes, (C) 0.5 Cu_2_O/PVDF DLHF membranes and (D) 0.75 Cu_2_O/PVDF DLHF membranes with outer dope flowrate 6 mL/min.

**Figure 9 membranes-12-00208-f009:**
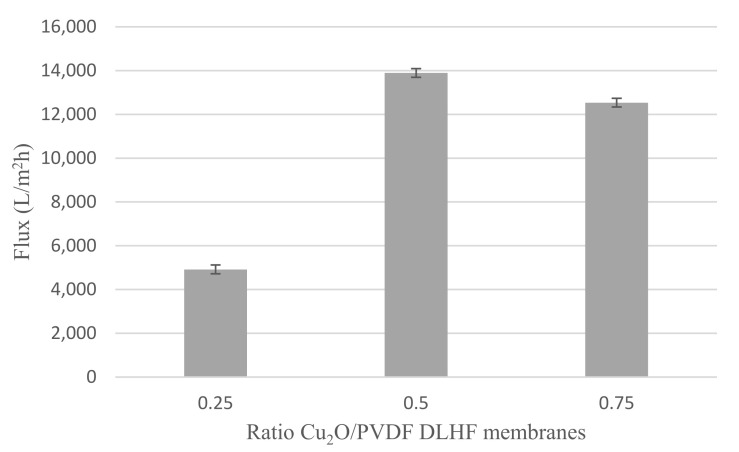
Water flux of 0.25, 0.5 and 0.75 Cu_2_O/PVDF DLHF membranes with outer dope flowrate 6 mL/min.

**Figure 10 membranes-12-00208-f010:**
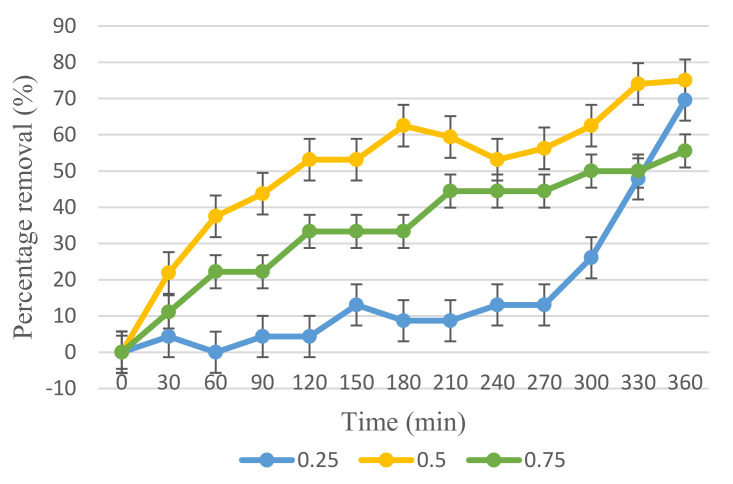
10 mg/L of BPA removal by Cu_2_O/PVDF DLHF membranes at different Cu_2_O/PDVF ratios: 0.25, 0.50 and 0.75 under visible light irradiation.

**Table 1 membranes-12-00208-t001:** Polymer dope solutions compositions.

Ratio	Outer Layer Composition (wt%)			Inner Layer Composition (wt%)		
Photocatalyst/Polymer	PVDF	Cu_2_O	DMAc	PVDF	PEG 6000	DMAc
0.25	15.0	3.75	81.25	15.0	3.0	82.0
0.50	15.0	7.5	77.5	15.0	3.0	82.0
0.75	15.0	11.25	73.75	15.0	3.0	82.0

**Table 2 membranes-12-00208-t002:** Outer layer thickness of 0.25 Cu_2_O/PVDF DLHF membranes for different outer layer dope flowrates: 3 mL/min, 6 mL/min and 9 mL/min.

Outer Layer Dope Flowrate (mL/min)	Outer Layer Thickness (nm)
3	13.05
6	21.35
9	89.35

**Table 3 membranes-12-00208-t003:** SEM image and EDX mapping of neat PVDF membrane and Cu_2_O/PVDF DLHF membrane with 6 mL/min outer dope flowrate.

Membrane	Cross Section	Copper
Neat PVDF membrane	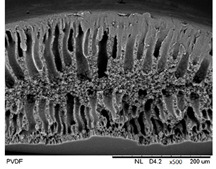	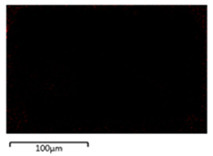
0.25 Cu_2_O/PVDF DLHF membrane	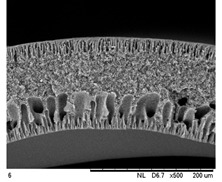	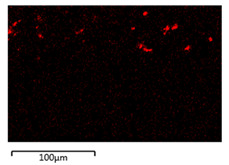
0.50 Cu_2_O/PVDF DLHF membrane	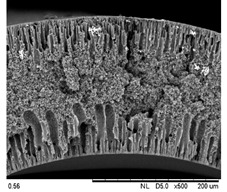	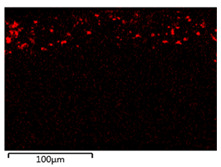
0.75 Cu_2_O/PVDF DLHF membrane	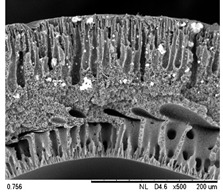	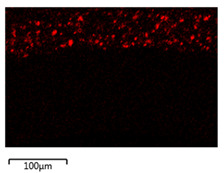

**Table 4 membranes-12-00208-t004:** AFM analysis of neat PVDF membrane and Cu_2_O/PVDF DLHF membrane.

Membrane	Membrane Surface	Surface Roughness (nm)
Neat PVDF membrane	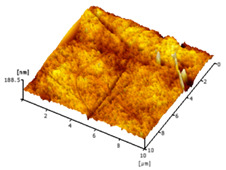	6.21
0.25 Cu_2_O/PVDF DLHF membrane	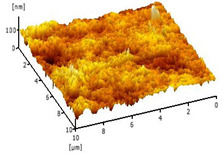	7.13
0.50 Cu_2_O/PVDF DLHF membrane	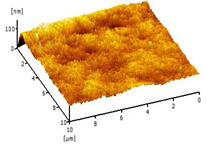	7.51
0.75 Cu_2_O/PVDF DLHF membrane	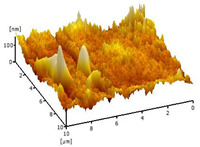	32.64

**Table 5 membranes-12-00208-t005:** Comparison between Cu_2_O/PVDF DLHF and other photocatalysts and BPA photodegradation.

Photocatalyst	UV/Visible Light	BPA Conc.	Removal (%)	References
N-doped TiO_2_/PVDF DLHF membrane	Visible light	5 mg/L	81.6%	[67]
Ag@TiO_2_ single layer hollow fiber membrane	Visible light	10 mg/L	90.51%	[68]
ZnO	UV light	25 mg/L	98%	[70]
TiO_2_	UV light	25 mg/L	65%	[70]
SnO_2_	UV light	25 mg/L	48%	[70]
CuNPs@CALB-3	UV light	45 mg/L	95%	[8]
WO_3_@MoS_2_/Ag hollow nanotubes	Visible light	10 mg/L	92.51%	[71]
WO_3_	Visible light	10 mg/L	10.55%	[71]
Visible light/Cu_2_O/H_2_O_2_	Visible light	10 mg/L	100%	[72]
Cu_2_O/PVDF DLHF membrane	Visible light	10 mg/L	75%	This research

## Data Availability

Not applicable.

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
