# Peer review of "Bisphenol A Removal Using Visible Light Driven Cu_2_O/PVDF Photocatalytic Dual Layer Hollow Fiber Membrane"

_membranes, 2022, doi:10.3390/membranes12020208_

Round 1
Reviewer 1 Report
The current manuscript described BPA removal using Cu2O-PVDF-DLHF membrane via visible light photocatalytic reaction. However, the work is interesting and can be useful for developing the HF membrane in water treatments there are some points that the authors should address before publishing:
Point 1: There are some spelling and editing errors such as extra spaces, singular and plural verbs, and nouns, the tenses and etc., that the MS needs to be double-checked carefully, lines 39-41, 43-44, 45, 51, 52, 53, 61, 66, 91, 348, and in the rest of paper.
Point 2: Line 135 what is the molecular weight of the PVDF?
Point 3: line 141, the condition for dissolving PVDF and the reason to use EtOH bath?
Point 4: Line 150-152 is vague.
Point 5: Line 266-268, it has been mentioned that “finger-like structures, 27 whereas the intermediate section of the membrane has a sponge-like structure” but in the discussion, there is no solid discussion about it, the author need to check some references and cite it as well. ( 10.1039/D0RA07592B, RSC Adv., 2020, 10, 40373-40383).
Point 6: The XRD miller indices to indicate the orientation of the plane needs to be determined.
Point 7: line 388, the effect of filler loading, why there is no evidence of neat PVDF membrane performance (0 wt. % of Cu2O)?
Point 8: Line 445, the hydrophilicity of the membrane first decreased then increased, Why?. In addition, the neat PVDF contact angle also needs to be reported for better comparison.
Point 9: line 469, normally adding the filler will boost the porosity of the membrane matrix, whilst in this section, the porosity decreased, why?
Point 10: line 497-501, what is mentioned, needs to be cited references.
Point 11: line 504-505, the few studies need to cite (https://doi.org/10.1016/j.chemosphere.2020.126966).
Point 12: The AFM results do not support the Contact Angle measurement, since roughness increases the surface area and increases in roughness leads to higher hydrophobicity. The Roughness increased while the C.A. decreased and then increases! Plus, the differences in the 0.25 and 0.75 Roughness are huge while the C.A. differences are not.
Author Response
Thank you for your review, Please see the attachment for the response.

Reviewer 2 Report
This article is devoted to the photo catalytic dual layer hollow fiber membrane fabrication. From point of view "membrane development" this article is article is well-done full cycle study. The fabrication technique is justified, morgology is studied, as well as photo catalytic properties. Average results are obtained (compare to literature). I think this article can be published in the Membranes. But there are some questions to authors.
- from this article it is not clear how obtained membranes will be working at real concentrations (below 1 micro g/l). In article there are initial concentrations about 5-10 mg/l. And, if it possible, the comparison with literature at the lower concentration can be made.
- Small remark: ethanol usually 96% (without further purification). Authors talk about "100 wt% ethanol". If it not mistake, the purity of initial ethanol or treatment processes should be mentioned.
Author Response
Thank you for your reviews. Please see the attachment for the responses.

Reviewer 3 Report
The authors aim at investigating the influence of outer layer dope flowrate on the characteristics of the Cu2O/PVDF membrane as well as the effect of Cu2O loading in the outer layer on membrane properties. The manuscript is overall well organized and structured with some interesting results, which would be of interest to the readers from BPA contaminated water treatment community. However, some concerns need to be addressed before acceptance.
- On what basis did the authors select a 10 mg/L concentration of BPA for membranes performance assessment?
- Why did the authors choose only three groups of data, i.e., the outer dope extrusion flowrate of 3 ml/min, 6 ml/min and 9 ml/min, for the study instead of more? Same problem for the ratio of Cu2O/PVDF (0.25, 0.50 and 0.75).
- It would be better to switch Tables 3 and 4 to Figures.
- Figure 10, error bars are needed to avoid any random data (possibly) to be used in the manuscript.
- Table 5, the authors are suggested to compare more reported data to show the superiority of the proposed membrane.
Author Response

(The authors gave the same response as above.)

Reviewer 4 Report
The comments concern mainly the introductory part.
Line 42 :Bisphenol A (BPA) or scientifically known as 2,2-bis-4-hydroxyphenylpropane (C15H16O2) consist of two methyl groups and one methyl bridge [3]
REMARK: 2,2-bis-4-hydroxyphenylpropane CORRECT: 2,2-bis-(4-hydroxyphenyl)propane
REMARK: Please insert DOI for the reference [3] DOI: 10.1007/s11356-014-3974-5
REMARK: BPA does not consist of two methyl groups and one methyl bridge. See below a text excerpt from the reference [3].
„Bisphenol A (2,2-bis(4-hydroxyphenyl)propane) is an organic compound composed of two phenol molecules bonded by a methyl bridge and two methyl groups (Table 1). BPA is used as an intermediate (binding, plasticizing, and hardening) in plastics, paints/lacquers, binding materials, and filling materials. Furthermore, it is used as an additive for flame-retardants, brake fluids, and thermal papers. About 95 % of BPA produced in industry is used to make plastics, in particular polycarbonate resins (71 %) and epoxy resins (29 %)
BPA is not produced naturally; it can be released into the environment during production and transport operations, from many products during their use or after their disposal in landfill, through effluent from wastewater treatment plants and from sewage sludge used in agriculture”
Line 73: Photocatalysis is a chemical reaction acceleration method that uses a catalyst and light REMARK It is not a scientific description of photocatalysis because photocatalysis is restricted to the acceleration of photoreactions.
Line 81: If the energy bandgap is equal to or greater than the energy bandgap COMMENT there is no logic in this statement
ADVICE: You can use information from the following article to improve the introductory paragraphs concerning photocatalysis: “Treatment of Produced Water with Photocatalysis: Recent Advances, Affecting Factors and Future Research Prospect”, published in an MDPI journal - Catalysts 2020, 10, 924; doi:10.3390/catal10080924
Line 97: Photocatalysts, DNA biosensors, sensors, cancer therapeutic agents, lithium-ion batteries, printed electronics, antimicrobial agents, and catalysis are just a few of the applications for Cu2O. REMARK Are you able to insert references separately for every application of Cu2O from this statement?
Line 116: PVDF is a form of polymer CORRECT: PVDF is a polymer
Line 116: REMARK: Biocompatible and bioneutral - Please cancel “bioneutral”
Line 124 Co-opting REMARK wrong use of this word
Line 124 Co-opting DLHF membrane technology with visible light photocatalyst BETTER: Dual Layer Hollow Fiber (DLHF) membrane technology with visible light photocatalyst. REMARK: ALL shortcuts must be explained at the first use
Line 126: There are few studies on Cu2O photocatalytic membrane. REMARK: If there are any studies, please add the references.
Line 156 Table 1. PEG 6000 REMARK: Not Polyethylene glycol 3000 (PEG, Sigma Aldrich, USA) , as in line 136?
Line 213: Atomic force microscopy (AFM) is a technique that uses air, liquid, or vacuum CORRECT Atomic force microscopy (AFM) is a technique that can be used in air, liquid, or vacuum
Line 301 work of Kamaludin et al. (2018) and Dzinun et al. (2015). REMARK References 31 and 32
Line 624: the photocatalyst 624 composition was varied between 0.25, 0.50, and 0.75 CORRECT the photocatalyst/polymer ratio in the composition was varied between 0.25, 0.50, and 0.75
QUESTION: Were the strength properties of the fibers acceptable with such a high Cu2O content?
Author Response

(The authors gave the same response as above.)

Round 2
Reviewer 1 Report
Most of the comments were addressed properly however the following comments need to be regulated:
A) What is discussed in comments number 12 and 9 must be included in the text since some observations in this study are not the usual observations. This will help the reader to have better insights into this work's observations.
B) In comment number 5, both Ref. need to be cited.
C) By molecular with it means by GPC (534000 or 275000, or...).
Author Response
Thank you for your commnets and reviews. Please see the attachment.
